# Decreasing Postoperative Pulmonary Complication Following Laparoscopic Surgery in Elderly Individuals with Colorectal Cancer: A Competing Risk Analysis in a Propensity Score–Weighted Cohort Study

**DOI:** 10.3390/cancers14010131

**Published:** 2021-12-28

**Authors:** Yih-Jong Chern, Jeng-Fu You, Ching-Chung Cheng, Jing-Rong Jhuang, Chien-Yuh Yeh, Pao-Shiu Hsieh, Wen-Sy Tsai, Chun-Kai Liao, Yu-Jen Hsu

**Affiliations:** 1Division of Colon and Rectal Surgery, Department of Surgery, Chang Gung Memorial Hospital, Linkou Branch, No. 5, Fu-Hsing St., Kuei-Shan, Taoyuan 33305, Taiwan; b9202063@adm.cgmh.org.tw (Y.-J.C.); you3368@adm.cgmh.org.tw (J.-F.Y.); b9302047@adm.cgmh.org.tw (C.-C.C.); chnyuh@cgmh.org.tw (C.-Y.Y.); hsiehps@cgmh.org.tw (P.-S.H.); wensy@cgmh.org.tw (W.-S.T.); mr9023@cgmh.org.tw (C.-K.L.); 2School of Medicine, Chang Gung University, Taoyuan 33305, Taiwan; 3Institute of Epidemiology and Preventive Medicine, National Taiwan University, Taipei City 10055, Taiwan; f05h41001@ntu.edu.tw

**Keywords:** colorectal cancer, elderly, laparoscopy surgery, outcome, propensity score, competing risk

## Abstract

**Simple Summary:**

As the effect of laparoscopic surgery on elderly patients with colorectal cancer (CRC) remains unclear, this propensity score–weighted cohort study revealed that laparoscopic surgery is a favorable method for elderly patients with CRC than open surgery in terms of less pulmonary-related postoperative morbidity and mortality, less hospital stay and similar oncological outcomes.

**Abstract:**

Advanced age is a risk factor for major abdominal surgery due to a decline in physical function and increased comorbidities. Although laparoscopic surgery provides good results in most patients with colorectal cancer (CRC), its effect on elderly patients remains unclear. This study aimed to compare the short- and long-term outcomes between open and laparoscopic surgeries in elderly patients with CRC. Total 1350 patients aged ≥75 years who underwent curative resection for stage I–III primary CRC were enrolled retrospectively and were divided into open surgery (846 patients) and laparoscopy (504 patients) groups. After propensity score weighting to balance an uneven distribution, a competing risk analysis was used to analyze the short-term and long-term outcomes. Postoperative mortality rates were lower in the laparoscopy group, especially due to pulmonary complications. Postoperative hospital stay was significantly shorter in the laparoscopy group than in the open surgery group. Overall survival, disease-free survival, and competing risk analysis showed no significant differences between the two groups. Laparoscopic surgery for elderly patients with CRC significantly decreased pulmonary-related postoperative morbidity and mortality in this large cohort study. Laparoscopic surgery is a favorable method for elderly patients with CRC than open surgery in terms of less hospital stay and similar oncological outcomes.

## 1. Introduction

The incidence of colorectal cancer (CRC) increases with age, and data from several populations indicate that approximately 40% of CRC cases occur in patients aged >75 years [1,2,3]. Surgery remains the mainstream treatment for elderly CRC. Some studies have reported that elderly patients undergoing open colorectal surgery are associated with high mortality (2.1–7.9%) and morbidity (17.7–50.0%) when compared with younger people [4,5,6]. Advanced age is associated with increasing comorbidities, such as diabetes mellitus, chronic kidney disease, and cardiovascular or pulmonary disease, which increase surgical morbidity and mortality.

Fortunately, laparoscopic surgery has progressed in the past two decades. Many randomized control trials, such as COST, CLASICC, COLOR, and COREAN have demonstrated that laparoscopic surgery has similar long-term outcomes and morbidity/mortality rates, but better short-term outcomes [7,8,9,10]. Less operative pain, less blood loss, shorter hospital stays, and shorter recovery times are the benefits of laparoscopic surgery, and it is an attractive choice for elderly patients. Although the operative risks of the potential cardiopulmonary change induced by pneumoperitoneum and the longer operative times were considered, several studies found better short-term outcomes in the laparoscopy for octogenarians [11,12,13]. Because elderly patients who undergo laparoscopic surgery can return to their regular lives faster, whether these advantages can further improve their long-term prognosis remains unclear Few prospective studies have compared laparoscopic surgery with open surgery in elderly patients, especially focusing on long-term outcomes [14].

This single-centered study aimed to compare the short-term postoperative complications and long-term oncological outcomes between open surgery and laparoscopic surgery in patients with CRC aged ≥75 years. Moreover, we conducted propensity score weighting analysis and competing risk analysis to adjust for possible biases impeding causal inference.

## 2. Materials and Methods

### 2.1. Patients and Variables

Detailed information regarding clinicopathological variables was retrieved from the Colorectal Section Tumor Registry of Chang Gung Memorial Hospital (CGMH). The Institutional Review Board of CGMH approved this study. Patient-related variables included age, sex, body weight, body height, body mass index (BMI), and underlying illness. Patients’ health information, such as incidences of hypertension, cardiac disease, cerebrovascular accident, asthma, diabetes mellitus, and liver cirrhosis, were collected. Previous surgical history, including appendectomy, cholecystectomy, hysterectomy, oophorectomy, and colorectal resection, were also collected. Blood analysis, including carcinoembryonic antigen (CEA), hemoglobin (Hb), albumin, aspartate aminotransferase (AST), total bilirubin, and creatinine (Cr), was performed before the operation. The tumor-related variables included tumor invasion depth (T stage), lymph node involvement (N stage), histologic subtype, histologic grade, tumor location, tumor size, and the number of retrieved lymph nodes.

Between January 2009 and December 2017, the patients aged ≥75 years who underwent curative radical resection for primary colorectal adenocarcinoma were enrolled in this study. The decision to perform laparoscopic surgery or open surgery depended on the physician’s and patient’s preferences. The Consort flow diagram is presented in Figure 1.

### 2.2. Short-Term Outcome and Long-Term Follow-Up

We measured short- and long-term outcomes. Short-term outcomes were postoperative morbidity and mortality, defined as surgical complications and death occurring within 30 days after surgery. Surgical complications included wound-related (infection or dehiscence), pulmonary (atelectasis or pneumonia), cardiovascular (myocardial infarction or stroke), bladder dysfunction, ileus, abdominal abscess, anastomosis (leakage or bleeding), and other rare complications including small bowel injury, enteritis, acute kidney injury, febrile non-hemolytic transfusion reaction (FNHTR), fluid overloaded, hypercalcemia, hyperparathyroidism, chylus ascites or combination of complications.

Physicians in the same department of this institute adopted similar follow-up routines and adjuvant treatment protocols. All patients participated in a follow-up program that included outpatient visits every 3 to 6 months for physical examination, CEA tests, chest radiography, abdominal sonography or abdominal computed tomography, and colonoscopy every 1 to 3 years postoperatively. The long-term outcomes were overall survival (OS), disease-free survival (DFS), and cause-specific mortality. We defined OS as the proportion of patients in the study who survived for a given period of time after receiving initial surgery. Disease-free survival was determined as the proportion of patients in the study who would not die and not recur for a given period of time after receiving initial surgery. The recurrence of cancer was confirmed by histology of biopsy specimens, re-operation, or radiological studies. The time to recurrence was defined as the duration between the date of the initial surgery and the recurrence confirmation date. Cause-specific mortality was defined as the proportion of patients who died from a specific cause for a given period of time after receiving initial surgery. We categorized the causes of death into two groups: patients who died from CRC and those who died from other causes.

### 2.3. Statistical Analysis

Differing baseline characteristics of the groups to be compared can confound bias and hinder causal inference in nonrandomized studies. Propensity scores [15,16] can be used to minimize such systematic differences. We estimated the propensity scores using logistic regression. Using propensity scores, we could obtain an unbiased estimate of the average treatment effect to determine whether laparoscopy could be applied to all patients in the study.

We applied propensity score weighting (PSW) [16,17] in the study. PSW generates a pseudo-population in which the treatment assignment is independent of the measured baseline covariates. The weights for each individual were calculated using the formula$\;w=\frac{z}{e}+\frac{1-z}{1-e}$, where z=1 to denote the laparoscopy group, z=0 to denote the open surgery group, and *e* denotes the estimated propensity score. The pseudo-population method minimizes systematic differences among the groups (as in a randomized study), consequently allowing for more robust causal inferences.

The overall survival and disease-free survival were calculated using the Kaplan–Meier method and compared using the PSW-adjusted log-rank test [18]. Crude and PSW-adjusted hazard ratios were estimated using Cox proportional hazards regression [19]. We also investigated whether patients in the laparoscopic group were more likely to die from colon cancer or die from other causes. A high competing risk of death occurs easily in a geriatric population with considerable comorbidities. Traditional approaches, including the Kaplan–Meier method and the Cox proportional hazards regression, can overestimate cause-specific mortality; therefore, we conducted a competing risk analysis [20,21]. The cause-specific mortality was calculated using the nonparametric cumulative incidence function estimator and compared using the PSW-adjusted Gray’s test [22]. The cause-specific hazard model [23] and subdistribution hazard model [24] were used to estimate the PSW-adjusted hazard ratio.

Categorical clinicopathological variables, presented as frequencies and proportions, were compared using the chi-square or Fisher’s exact test. The Wilcoxon rank-sum test was used to analyze continuous variables, expressed as medians and ranges. Statistical significance was set at *p* < 0.05. All statistical analyses were performed using SAS (version 9.4; SAS Institute, Cary, NC, USA).

## 3. Results

We enrolled 1350 patients who underwent CRC surgery. Of these, 846 patients received conventional open surgery, and 504 patients underwent laparoscopic surgery. Table 1 shows the baseline characteristics of patients in the open group versus the laparoscopy group before and after propensity score weighting. Before propensity score weighting, the laparoscopic group had a greater proportion of patients with higher BMI, less previous colorectal surgery, comorbidity of hypertension, preoperative carcinoembryonic antigen < 5 ng/mL, serum albumin level > 3.5 mg/dL, early tumor stage, well-differentiated histologic grade, retrieved lymph node > 12, and smaller tumor size compared with the open group. After PSW, the two groups did not significantly differ in any patient characteristics, except that the open group had more previous abdominal surgery for colorectal surgery.

Table 2 shows the postoperative outcomes of patients in the open group versus those in the laparoscopy group before and after PSW. Before PSW, the two groups had no significant difference in postoperative mortality (including lung, cardiovascular event, abdomen, and anastomosis related). After PSW, the laparoscopic group had a lower postoperative mortality rate (open vs. laparoscopic, 1.89% vs. 0.79%, *p* = 0.033) and a lower rate of mortality induced by pulmonary complications (open vs. laparoscopic, 1.06% vs. 0.20%, *p* = 0.004). Pulmonary-induced mortality accounted for a high proportion (9/16, 56.3%) in all mortalities after open surgery, and only one patient (1/4: 25%) died from pulmonary reasons after laparoscopic surgery. The open and laparoscopic groups had similar overall postoperative morbidity rates before and after PSW. In view of the subclassifications of morbidities, after PSW, the laparoscopic group had higher rate of abdominal abscess and anastomosis-related complications (open vs. laparoscopic, abdomen: 0.71 vs. 1.59, *p* = 0.015; anastomosis: 0.83% vs. 3.17%, *p* = 0.001). The postoperative hospital stay was significantly shorter in the laparoscopic group both before and after PSW.

Table 3 shows the results of the estimated hazard ratio of patients in the open group versus those in the laparoscopy group. The PSW-adjusted hazard ratios were 1.069 (overall survival) and 1.081 (disease-free survival) in the two groups, and these weak effects were not statistically significant. The two groups also had similar disease-free survival curves (*p* = 0.39, Figure 2A) and overall survival curves (*p* = 0.50, Figure 2B). Table 4 shows the results of the estimated cause-specific hazard ratio and subdistribution hazard ratio of patients in the open group versus those in the laparoscopy group. The PSW-adjusted cause-specific hazard ratio for the two groups was 0.966 for patients who died from colon cancer and 1.123 for patients who died from other causes, and these weak effects were not statistically significant. The PSW-adjusted subdistribution hazard ratio for the two groups was 0.925 for patients who died from colon cancer and 1.074 for patients who died from other causes. However, these weak effects were not statistically significant. The two groups also had similar cause-specific mortality curves for patients who died from CRC (*p* = 0.84, Figure 3A) and patients who died from other causes (*p* = 0.33, Figure 3B).

## 4. Discussion

This is the largest retrospective study from a single academic medical center analyzing laparoscopic radical surgery for elderly CRC patients using PSW to balance the uneven distribution and competing risk analysis for elderly patients. For elderly patients with non-metastatic CRC who underwent radical resection, our report revealed a similar overall postoperative morbidity rate between the open and laparoscopic groups. In addition, the postoperative mortality rates were lower in the laparoscopy group, especially due to pulmonary complications. The postoperative hospital stay was significantly shorter in the laparoscopic group both before and after PSW.

As a retrospective study from a single medical institute, even though we provide, to our knowledge, the largest sample size to analyze the efficiency of laparoscopic surgery for elderly CRC patients, the bias between the open and laparoscopic groups existed for several reasons, including physicians and patient’s preferences. Hence, we applied PSW to correct the bias and keep all cases in roll rather than deleted cases, such as the propensity score matching (PSM) method. The method of PSW may present the originality of this cohort. Our data showed no significant differences in several confounding factors that may affect short-term and long-term outcomes after PSW.

In this study, the laparoscopy group had a lower postoperative mortality rate than the open group. Some studies showed that there was no significant increase in mortality rate in the open group [11,12,25,26,27,28,29,30,31] and the results were similar to our data before PSW. After the application of PSW, the mortality rate was significantly lower in the laparoscopic group. The use of PSW allowed us to adjust for multiple confounders simultaneously. This may increase the differences between the observed and expected values of the chi-squared statistics, resulting in a higher tendency of significance. The inconsistent results between studies may be due to the heterogeneity of the different cohorts, the operation skill, facility improvement, or unmeasured confounding bias. In addition, mortality caused by pulmonary complications was significantly higher in the open group (up to 56.3% of all mortality cases). Pulmonary complications, such as atelectasis and pneumonia may be caused by the poor pulmonary toilet, and profound laparotomy wound pain would lead to diaphragmatic dysfunction and reduced chest ventilation after surgery [12]. In the elderly, studies have reported that postoperative pulmonary complications are an important cause of postoperative mortality [32,33]. Postoperative pulmonary complications were also higher in the open group after PSW in this study, and the results were also presented in several studies [12,26,34]. Laparoscopic colorectal resection may be an appropriate selection for elderly patients because a smaller wound decreases the rate of fatal pulmonary complications.

In the elderly, anastomotic leakage after colorectal surgery is also an important concern. They may present comorbidities, relatively poor nutritional status, and lower activity of daily living status, which may contribute to an increased rate of anastomotic leakage. Hence, surgeons prefer stoma creation (diverting stoma or end-stoma) in the elderly compared to younger populations [25]. Postoperative abdominal complications and anastomotic leakage were higher in the laparoscopic group after PSW in this study. This result is different from other studies that showed no difference in the anastomotic leakage rate between the open and laparoscopic groups [12,29–31,35). The first reason to explain this result is that the anastomotic leakage rate in the open group was much lower (0.83%) than that in similar studies (2.6–8.4%) [11,25,27,29,31,34,35,36]. When the selection of elderly patients to receive stoma creation is precise and appropriate, it will lead to avoidance of anastomosis leakage-related symptoms and signs. A diverting stoma may mask the symptoms of a leak, resulting in an incorrect estimation of the leakage rate. The stoma rate in the open group was higher, which may have caused a significantly lower anastomosis rate than that in the laparoscopic group. Second, in the early era of laparoscopy application, the instruments were not very advanced, and the old-type stapler for anastomosis had a higher rate of anastomosis insufficiency [37]. Third, the blood supply to the anastomosis stump may not be easy to confirm through laparoscopy in the past. Currently, improvement of the stapler and the use of ICG to confirm the blood supply to the anastomosis will decrease the anastomosis rate. The anastomosis leakage rate (3.17%) in the laparoscopic group in our study was not higher than that in other studies (0–5.9%) [11,25,27,29,31,34,35,36,38]. Our previous study revealed a higher rate of anastomotic leakage in laparoscopic surgery because of the lower rate of stoma creation when compared with the open group [39]. In our practice, in laparoscopic surgery, under adequate postoperative monitoring and care strategy, even anastomosis leakage attacked, the mortality rate did not increase in our results. After the leakage occurred after laparoscopic surgery, conservative treatment with or without diverting stoma creation, abdominal and pelvic cavity irrigation through laparoscopy (only trocar wounds with a lower infection rate), or drainage abscess by image-guide were the main treatments in our hospital.

The elderly cancer patients had higher rates of death from non-cancer causes, such as cardiovascular and cerebrovascular accidents, and serious infections during follow-up, which may lead to underestimation of the cancer-related risk. We used PSW to minimize bias between the two groups and conducted a competing risk analysis for these elderly patients. The PSW-adjusted hazard model showed that OS and DFS were not significantly different between the open and laparoscopic groups. For OS and DFS, several studies have reported no significant differences in long-term outcomes between the open and laparoscopic groups [11,29,40,41]. In the competing risk analysis, our results showed that in the cause-specific hazard model, for patients who died from colorectal cancer or other causes, the weak effects were not statistically significant. Two studies have analyzed cancer-specific survival in elderly patients with CRC, and the results showed similar outcomes between the open and laparoscopic groups [41,42]. Shigeta et al. analyzed 107 elderly CRC patients with competing-risk regression analysis, which reported no significant association between the surgical procedure and the three types of death (cancer-related death, cardiopulmonary death, and other deaths) [35]. This study showed that laparoscopic resection for elderly patients with CRC had similar oncologic long-term outcomes compared with open resection by extensive statistical approaches. The results presented real-world experience in a single institute to strengthen the laparoscopic procedure for elderly patients with CRC.

This study had several limitations. This is a single-center retrospective analysis. This retrospective study still had selection bias, although PSW was used to reduce the imbalance. In addition, this study is a long-period cohort that crosses approximately 9 years. During this period, the laparoscopic instruments and anastomosis stapler progressed and advanced, which may have caused bias in this study. In this study, the learning curve might influence the outcomes on both open and laparoscopic groups that we did not analyze.

## 5. Conclusions

Laparoscopic radical resection for elderly patients with CRC significantly decreased pulmonary-related postoperative morbidity and mortality in this large study, which conducted propensity score weighting to minimize the bias and competing risk analysis for the elderly. Moreover, laparoscopic surgery had the advantage of less hospital stay and similar oncological outcomes including overall survival and disease-free survival as those of open surgery. Laparoscopic radical resection is a favorable method for elderly patients with CRC.

## Figures and Tables

**Figure 1 cancers-14-00131-f001:**
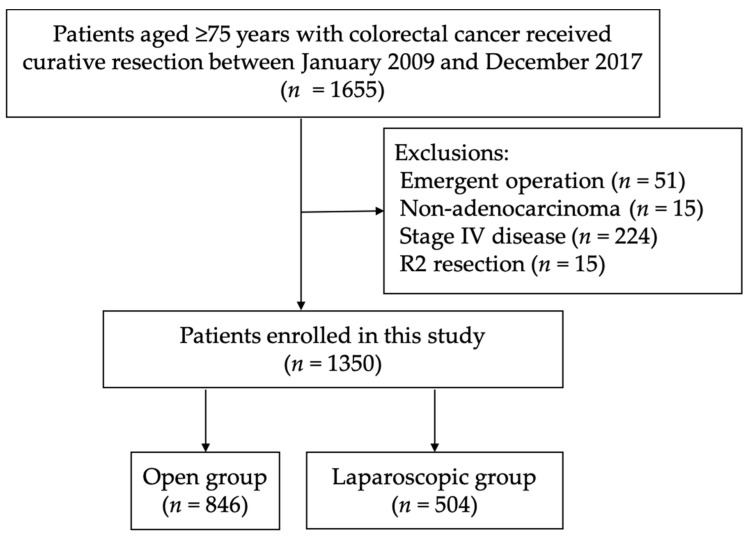
The Consort flow diagram of this study.

**Figure 2 cancers-14-00131-f002:**
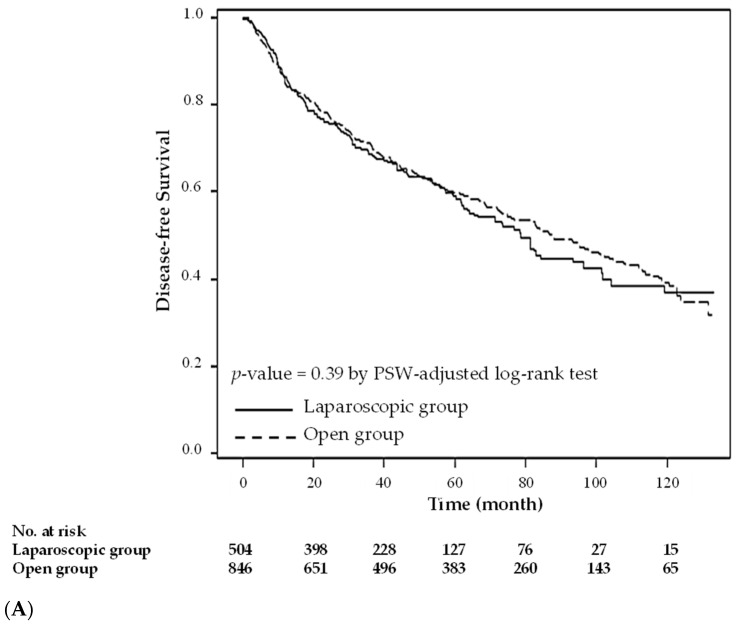
Overall survival and disease-free survival after propensity score weighting. (**A**). Disease-free survival; (**B**). Overall survival.

**Figure 3 cancers-14-00131-f003:**
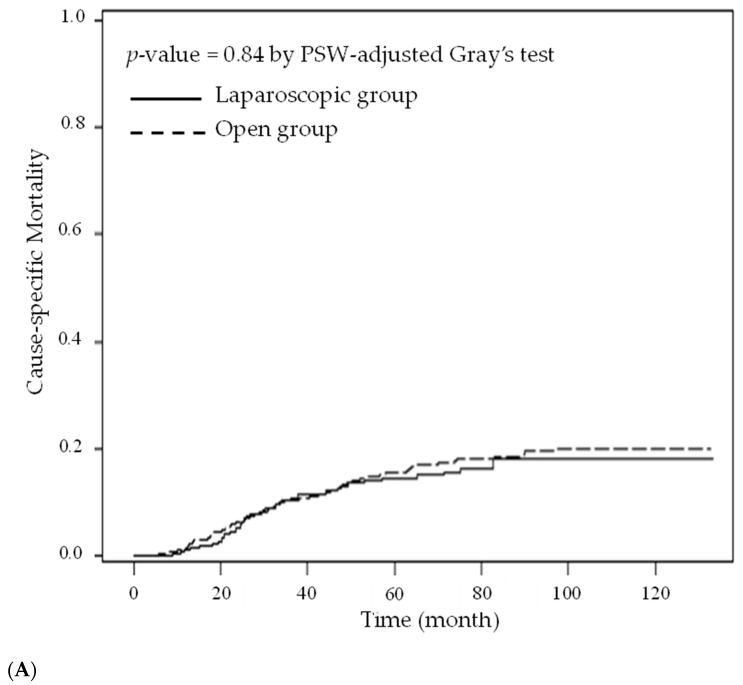
Cause-specific mortality after propensity score weighting. (**A**). Death from this cancer; (**B**). Death from other causes.

**Table 1 cancers-14-00131-t001:** Baseline characteristics of patients treated with open surgery vs. laparoscopy before and after propensity score weighting.

Variables	Open Group	Laparoscopic Group	*p*-Value	
	*n* = 846	*n* = 504	Before	After
Age, No. (%)			0.85	0.95
75–80 years	415 (49.05)	250 (49.60)		
≥80 years	431 (50.95)	254 (50.40)		
Gender, No. (%)			0.73	0.92
Female	378 (44.68)	230 (45.63)		
Male	468 (55.32)	274 (54.37)		
BMI, No. (%)			<0.01	0.75
<25	596 (70.45)	315 (62.50)		
≥25	250 (29.55)	189 (37.50)		
Previous abdominal operation, No. (%)				
Appendectomy	97 (11.47)	54 (10.71)	0.67	0.02
Cholecystectomy	59 (6.97)	36 (7.14)	0.91	0.01
Hysterectomy	50 (5.91)	20 (3.97)	0.12	0.09
Oophorectomy	14 (1.65)	6 (1.19)	0.49	0.03
Colon-rectal operation	44 (5.20)	12 (2.38)	0.01	0.10
total	584 (69.03)	336 (66.67)	0.37	<0.01
Operation, No. (%)				
Hartmann resection	38 (4.49)	16 (3.17)	0.23	0.59
Abdomino-peritoneal	18 (2.13)	14 (2.78)	0.45	0.08
Anterior resection	473 (55.91)	295 (58.53)	0.35	0.92
Left hemicolectomy	55 (6.50)	35 (6.94)	0.75	0.17
Right hemicolectomy	225 (26.60)	139 (27.58)	0.69	0.48
Segmental resection	18 (2.13)	2 (0.40)	0.01	<0.01
Subtotal colectomy	19 (2.25)	3 (0.60)	0.02	<0.01
Comorbidity, No. (%)				
Hypertension	479 (56.62)	327 (64.88)	<0.01	0.89
Cardiac disease	151 (17.85)	90 (17.86)	1.00	0.99
Cerebrovascular accident	70 (8.27)	32 (6.35)	0.20	0.59
Asthma	39 (4.61)	23 (4.56)	0.97	0.91
Diabetes mellitus	214 (25.30)	131 (25.99)	0.78	0.98
Liver Cirrhosis	96 (11.35)	56 (11.11)	0.89	0.98
others	277 (32.74)	186 (36.90)	0.12	0.81
Carcinoembryonic antigen, No. (%)			<0.01	0.65
<5 ng/mL	537 (63.48)	365 (72.42)		
≥5 ng/mL	309 (36.52)	139 (27.58)		
Hemoglobin, No. (%)			1.00	1.00
<10 mg/mL	225 (26.60)	134 (26.59)		
≥10 mg/mL	621 (73.40)	370 (73.41)		
Albumin, No. (%)			<0.01	0.78
<3.5 mg/dL	200 (23.64)	82 (16.27)		
≥3.5 mg/dL	646 (76.36)	422 (83.73)		
Total bilirubin, No. (%)			0.78	0.77
≤1.3	831 (98.23)	494 (98.02)		
>1.3	15 (1.77)	10 (1.98)		
Creatinine, No. (%)			0.55	0.75
≤1.3	709 (83.81)	416 (82.54)		
>1.3	137 (16.19)	88 (17.46)		
Tumor stage, No. (%)			<0.01	0.96
1	111 (13.12)	100 (19.84)		
2	369 (43.62)	224 (44.44)		
3	366 (43.26)	180 (35.71)		
Histological type, No. (%)			0.36	0.60
Adenocarcinoma	794 (93.85)	479 (95.04)		
Mucinous adenocarcinoma & Signet ring cell	52 (6.15)	25 (4.96)		
Histology grade, No. (%)			0.01	0.80
Poorly differentiated	85 (10.05)	30 (5.95)		
Moderately differentiated	682 (80.61)	412 (81.75)		
Well differentiated	79 (9.34)	62 (12.30)		
Retrieved lymph node (+) number, No. (%)			0.01	0.18
<12	47 (5.56)	13 (2.58)		
≥12	799 (94.44)	491 (97.42)		
Tumor site, No. (%)			0.52	0.91
Left side colon	325 (38.42)	182 (36.11)		
Anus & rectum	266 (31.44)	173 (34.33)		
Right side colon	255 (30.14)	149 (29.56)		
Tumor size, No. (%)			<0.01	0.99
<4 cm	422 (49.88)	297 (58.93)		
≥4 cm	424 (50.12)	207 (41.07)		
Stomy type, No. (%)			0.07	0.96
No	671 (79.31)	425 (84.33)		
Diverting stomy	103 (12.17)	46 (9.13)		
End stomy	72 (8.51)	33 (6.55)		

**Table 2 cancers-14-00131-t002:** Postoperative outcomes of patients treated with open surgery vs. laparoscopy before and after propensity score weighting.

Variables	Open Group	Laparoscopic Group	*p*-Value	
	*n* = 846	*n* = 504	Before	After
Postoperative mortality, No. (%)	16 (1.89)	4 (0.79)	0.106	0.033
Pulmonary	9 (1.06)	1 (0.20)	0.071	0.004
Cardiovascular event	3 ^+^ (0.35)	1 (0.20)	0.610	0.646
Abdominal abscess	1 (0.12)	0 (0.00)	0.440	0.220
Anastomosis	1 (0.12)	1 (0.20)	0.711	0.306
Others	2 (0.24)	1 (0.20)	0.886	0.657
Postoperative morbidity, No. (%)	159 (18.79)	82 (16.27)	0.241	0.218
Wound	41 * (4.85)	14 (2.78)	0.063	0.061
Pulmonary	30 (3.55)	9 (1.79)	0.062	0.003
Cardiovascular event	3 ^+^ (0.35)	2 ^++^ (0.40)	0.902	0.565
Bladder dysfunction	23 (2.72)	10 (1.98)	0.398	0.224
Ileus	30 (3.55)	15 (2.98)	0.573	0.373
Abdominal abscess	6 (0.71)	8 (1.59)	0.124	0.015
Anastomosis	7 ^#^ (0.83)	16 ^##^ (3.17)	0.001	0.001
Others	19 (2.25)	8 (1.59)	0.403	0.748
Clavien-Dindo Classification, No. (%)			0.391	0.378
Grade I, II	113 (13.36)	54 (10.71)		
Grade III, IV, V	46 (5.44)	28 (5.56)		
Conversion, No. (%)	-	8 (1.59)	-	-
Length of hospital stay, median (IQR), day	10.5 (9–15)	8 (7–11)	<0.001	<0.001

* four patients had wound dehiscence; ^+^ two patients were acute myocardial infarction, and one patient was stroke; ^++^ two patients were acute myocardial infarction; ^#^ all the seven patients were anastomosis leakage; ^##^ two patients had anastomosis bleeding, the other 14 patients were anastomosis leakage.

**Table 3 cancers-14-00131-t003:** Results of the estimated hazard ratio of patients treated with open surgery vs. laparoscopy.

Operation Methods	Disease-Free Survival		Overall Survival	
	Crude HR [95% CI]	PSW-Adjusted HR [95% CI]	Crude HR [95% CI]	PSW-Adjusted HR [95% CI]
Laparoscopic group	0.939 [0.789–1.117]	1.081 [0.965–1.211]	0.940 [0.781–1.132]	1.069 [0.946–1.207]
Open group	-	-	-	-

Abbreviations: HR, hazard ratio; CI, confidence interval; PSW, propensity score weighting.

**Table 4 cancers-14-00131-t004:** Results of competing risk analysis of patients treated with open surgery vs. laparoscopy.

Operation Methods	Cause-Specific Hazard Model		Subdistribution Hazard Model	
	Death from This Cancer PSW-Adjusted HR [95% CI]	Death from Other Causes PSW-Adjusted HR [95% CI]	Death from This Cancer PSW-Adjusted HR [95% CI]	Death from Other Causes PSW-Adjusted HR [95% CI]
Laparoscopic group	0.966 [0.782–1.194]	1.123 [0.968–1.303]	0.925 [0.750–1.141]	1.074 [0.927–1.244]
Open group	-	-	-	-

Abbreviations: HR, hazard ratio; CI, confidence interval; PSW, propensity score weighting.

## Data Availability

The data presented in this study are available on request from the corresponding author.

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
