# Peer review of "Decreasing Postoperative Pulmonary Complication Following Laparoscopic Surgery in Elderly Individuals with Colorectal Cancer: A Competing Risk Analysis in a Propensity Score–Weighted Cohort Study"

_cancers, 2021, doi:10.3390/cancers14010131_

Round 1

Reviewer 1 Report

Dear Authors,

I would like to congratulate you with this fine work.

However, I have some comments.

INTRODUCTION

-You cite a very old date in ref 1-3. These must me updated, as most of the patients are diagnosed with CRC between the 50 and 75, and the numbers are increasing in a subgroup up to 50 age of years.

-Ref 4-6 are also at least 10 years of old. There are much more recent papers. 

-"Few prospective studies have compared laparo-55 scopic surgery with open surgery in elderly patients, especially focusing on long-term 56 outcomes" [ref]

MATERIALS

I am confused: in the intro part you aimed to compare the outcomes in patients over 75 years, however, in the results section Table 1 you are comparing <80 and > 80. 

You also write that you have included 75 patients, however in the results section the number is 1350.

You should also add the Consort flow diagram for included/excluded patients.

So how many patients have been actually included in the study? How many of them left following PSW?

Complications must be classified according to the Clavien Dindo. 

What was the standard preparation of the patients for surgery?

Was the ERAS guidelines used?

RESULTS

Table 2 - the names of the graphs must be explain. What does it mean Abdomen? Anastomosis? Others? Differentiation between the leak and stenosis is crucial. How did you define the leak? the abscess? differentiation between the wound infection vs dehiscence is crucial. Same goes with all the explanations of the complications. 

You must also included what kind of operations were performed. 

How many elective surgery vs emergency in both groups?

DISCUSSION

Limitation section - single center, retrospective analysis. 

Reviewer 2 Report

The study by Chern and colleagues is one of the largest retrospective series comparing short and long term outcome between laparoscopic and open surgery in elderly patients with CRC. Using a competing risk analysis in a propensity score-weighted population, the investigators found lower postoperative mortality rates (due to less pulmonary complications) and shorter postoperative hospital stay in the laparoscopic group, while OS and DFS were not different between both groups. Based on their finding, the investigators conclude that laparoscopic surgery is the favorable surgical approach in elderly CRC patients.

This study has generated important and clinically relevant results that are of interest to the scientific community/readers of Cancers. The following comments should be addressed:

Major comments:

  1. As this is a retrospective analysis, the availability of detailed shared decision-making is crucial to avoid selection bias. Because it is mentioned that the choice depended on “the physician’s and patient’s preferences”, it would be informative to include the most important considerations from both sides that have led to the choice of surgical approach.
  2. Can the authors comment on the professional experience of the team to perform the laparoscopic resection? Was there a learning curve between 2009 and 2017? Have the surgical techniques been modified during this period?
  3. An important additional reason to consider the laparoscopic approach is the cost-effectiveness as compared to the open technique. The authors should discuss this aspect as well.

Minor comments:

  1. Abstract. No numbers of patients enrolled in the analyses are mentioned. This information should be provided already in this section of the manuscript.
  2. Page 2, line 78: the number of 75 patients who “underwent (…) were enrolled in this study”, is confusing. Please provide the appropriate numbers or explain this 75.
  3. Conclusion (page 10); please specify the absence of a difference in OS/EFS also here.

Round 2

Reviewer 1 Report

Dear Authors, 

The manuscript improved a lot.